# Synthesis of Novel Tetra(µ_3_-Methoxo) Bridged with [Cu(II)-O-Cd(II)] Double-Open-Cubane Cluster: XRD/HSA-Interactions, Spectral and Oxidizing Properties

**DOI:** 10.3390/ijms21228787

**Published:** 2020-11-20

**Authors:** Abderrahim Titi, Mouslim Messali, Rachid Touzani, Mohammed Fettouhi, Abdelkader Zarrouk, Nabil Al-Zaqri, Ali Alsalme, Fahad A. Alharthi, Amjad Alsyahi, Ismail Warad

**Affiliations:** 1Laboratory of Applied and Environmental Chemistry (LCAE), Mohammed First University, 60000 Oujda, Morocco; aboutasnim@yahoo.fr (M.M.); r.touzani@ump.ac.ma (R.T.); 2Department of Chemistry, King Fahd University of Petroleum and Minerals, P.O. Box 5048, Dhahran 31261, Saudi Arabia; fettouhi@kfupm.edu.sa; 3Laboratory of Materials, Nanotechnology and Environment, Faculty of Sciences, Mohammed V University, Av. I bn Battouta, Box 1014, 47963 Agdal-Rabat, Morocco; azarrouk@gmail.com; 4Department of Chemistry, College of Science, King Saud University, P.O. Box 2455, Riyadh 11451, Saudi Arabia; nalzaqri@ksu.edu.sa (N.A.-Z.); aalsalme@ksu.edu.sa (A.A.); fharthi@ksu.edu.sa (F.A.A.); 438203007@student.ksu.edu.sa (A.A.); 5Department of Chemistry, Science College, An-Najah National University, P.O. Box 7, Nablus 00970, Palestine

**Keywords:** Cd-O-Cu cluster, XRD/HSA, catecholase, spectral

## Abstract

A new double-open-cubane core Cd(II)-O-Cu(II) bimetallic ligand mixed cluster of type [Cl_2_Cu_4_Cd_2_(NNO)_6_(NN)_2_(NO_3_)_2_].CH_3_CN was made available in EtOH/CH_3_CN solution. The 1-hydroxymethyl-3,5-dimethylpyrazole (NNOH) and 3,5-dimethylpyrazole (NNH) act as N,O-polydentate anion ligands in coordinating the Cu(II) and Cd(II) centers. The structure of the cluster in the solid state was proved by XRD study and confirmed in the liquid state by UV-vis analysis. The XRD result supported the construction of two octahedral and one square pyramid geometries types around the four Cu(II) centers and only octahedral geometry around Cd(II) two centers. Interestingly, NNOH ligand acts as a tetra-µ^3^-oxo and tri-µ^2^-oxo ligand; meanwhile, the N-N in NNH acts as classical bidentate anion/neutral ligands. The interactions in the lattice were detected experimentally by the XRD-packing result and computed via Hirschfeld surface analysis (HSA). The UV-vis., FT-IR and Energy Dispersive X-ray (EDX), supported the desired double-open cubane cluster composition. The oxidation potential of the desired cluster was evaluated using a 3,5-DTB-catechol 3,5-DTB-quinone as a catecholase model reaction.

## 1. Introduction

Multinuclear cluster chemistry has gained high interest due to their biochemistry potential applications and structural diversity [1]. The synthesis of multinuclear Cu(II) complexes have been planned via various factors, for example, counter ions, ligands, reagents sequence, solvents, temperature and pH [2]. Cu(II) complexes with various ligands compositions and polynuclear structures are recorded in the literature with their molecular biology, magnetism applications and catalysis. Multi-copper clusters can enhance several oxidation reactions of amines and alcohols [3,4,5,6,7]. Copper (II) complexes clusters can also serve as metallo-pharmaceutical agents as antitumor, antimicrobials, antibacterial, antifungal, antipyretic, antidiabetic and antiviral agents [8,9,10,11,12]. The use of O, N, P-ligands as polydentate was been a good preference to build multilateral and multinuclear architectures cluster because of their electronic, steric effect and several coordination modes [13,14,15,16]. Tetra-nuclear Cu(II) clusters are existing as hot-urge points in bioinorganic modeling, magnetochemistry, multielectron transfer and catalysis. In the two last decades, plenty of cubane clusters were prepared by using alkoxo-bridged NOO or NNO types of donor ligands were prepared until now [17,18,19,20,21,22].

On the other hand, clusters with multiple spin molecular centers revealed a good catalytic aspect, especially in industrial oxidation processes [23,24,25,26,27,28,29]. Clusters with chelate mixed alcoholic pyrazole ligands and metal have recently received less interest due to their difficulty in preparation, low stability and rear in collecting suitable crystals to be judged by XRD single structure analysis [25,26,27,28].

We recently synthesized several novel tetra-nuclear metal cubane clusters; their structures were evaluated by XRD analysis, and their cluster catecholase catalytic activities were also evaluated by converting catechol to O-quinone as an oxidation model [21,22,30]. In this piece of work, the bimetallic Cd-O-Cu double-open cubane [Cl_2_Cu_4_Cd_2_(NNO)_6_(NN)_2_(NO_3_)_2_].CH_3_CN cluster made available using chelate NNOH and NNH ligands, the structure of the cluster was proven by XRD, the tetrahedral-*µ*_3_-O and trigonal pyramidal-*µ*_2_-O bridges were detected. The octahedral and square pyramid geometries were resolved for both the metal centers. Moreover, physicochemical and HS analyses were determined to ensure the catecholase catalytic process of the desired cluster in mild or harsh conditions.

## 2. Results and Discussion

### 2.1. Cluster Preparation

The desired bimetallic cluster was prepared by stirring CuCl_2_.2H_2_O and Cd(NO_3_)_2_.4H_2_O metallic salts together with 1-hydroxymethyl-3,5-dimethylpyrazole (NNOH) and 3,5-dimethylpyrazole (NNH) at RT for 24 h using EtOH:CH_3_CN solution (Scheme 1). The desired reaction of clusterization was performed at RT in an open O_2_ atmosphere with equivalent amounts of each ligand and metal salt, and the final bimetallic [Cl_2_Cu_4_Cd_2_(NNO)_6_(NN)_2_(NO_3_)_2_].CH_3_CN cluster was isolated in 78% yield. Moreover, the 3D structure was definite by XRD analysis (for the first time).

### 2.2. Single Crystal X-ray Diffraction (SC-XRD) Investigation

The 3D structure of the newly desired cluster is shown in Figure 1, whereas the selected atomic and angles distances are provided in Table 1. The cluster crystallized in the monoclinic/*P*21/*c* crystal system and space group, respectively. The cluster crystalized with 4Cu and 2Cd metal ion double open cubane core centers with [Cl_2_Cu_4_Cd_2_(NNO)_6_(NN)_2_(NO_3_)_2_].CH_3_CN formula. All the organic NNOH and NNH and inorganic NO_3_ and Cl ligands acted as chelate or bridge anion donors, which stabilized the cluster as neutral with no, counter ions (Figure 1a). No solvents like MeOH or water molecules were detected, but only one uncoordinated CH_3_CN molecule was present in the crystal lattice (Figure 1b). The cationic units of the two Cd(II) and four Cu(II) centers were not directly bonded. On the other hand, all the centers had methoxo-bridge like 4μ^3^-O, 2μ^2^-O, 2μ^2^-NO_3_ or 2μ^2^-Cl- bridge functional groups. Furthermore, both NN^−^ ligands, which acted as terminal bridge donors bidentate that coordinated to the Cu(II) centers only, supported the formation of two different Cu(II) geometrical centers: terminal 2Cu(II) centers with 6-coordinate [2N, 3O, 1Cl], constructing an octahedral center. Both centers were saturated with one OՈO-bidentate NO_3_ ligand (Figure 1c). Conversely, the other Cu(II) core centers were sterically forced to be with 5-coordinate [2N, 3O], constructing a square pyramid geometry center with τ_[1O2O6N1N]_ = 3.12 °C (Figure 1d). Both core 2Cd(II) centers were constructed with distorted octahedral centers once the 6-coordinates [1N, 4O, 1Cl] were recorded (Figure 1e). It was delightful to observe that NNO^−^ ligands coordinated with the both Cu and Cd centers via η^3^:η^1^-O,N-modes of coordination, hence the di-μ^2^-methoxo-trigonal pyramid-O center (Figure 1f), and two different tetra-μ^3^-methoxo-tetrahedral-O centers (Figure 1g,h).

Experimentally, several polar shorter interactions < 3 Å were detected in the crystal lattice since the cluster contains O, N and Cl heteroatoms together with polar H atoms. The non-bonded O atom of NO_3_ ligands played a critical role in the building of the net of H-bonds in cluster lattice. Therefore, 4 × 2 H_CH_….O-_NO2_ with 2.58 and 2.70 Å (Figure 2a) with full geometric parameters (Å, degree) are illustrated in Table 2 are recorded. The C-H…..πC=C ring of NN ligands as a short interaction with 2.80 Å distance played a critical role in stabilizing the crystal lattice since four bonds of such type were recorded (Figure 2b) and the interesting solvent interactions 2Cl…. Π CN_CH3CN_ with 3.56 Å (Figure 2c) [31,32,33,34,35].

### 2.3. HS and 2D-FP Investigation

The surface was mapped to obtain more interaction information on the molecule and its surrounding molecules in the crystal lattice that played critical roles in stabilizing the structural formula via short intermolecular forces reflected by red spot sizes on the normalized d_norm_ [36,37,38,39,40,41]. The results illustrated in Figure 3 showed that the values of HS with three-dimensional shape and cave centers and shape ranged from 0.723 to 1.874 a.u. The red spots were behind the polar atoms such as oxygen, nitrogen, chlorine and hydrogen. Eleven red points were detected on the computed cluster surface, and they were attributed to the existence of 8 H-bond interactions. Two C-H…..O-NO_2_ (Figure 3a–c) formed H-bonds with a distance of 2.56 and 2.72 Å and two C_ring_-H….C=C with a length of 2.76 Å (Figure 3d). The computed HS and experimental XRD interactions were highly matched in their positions and structural parameter values.

The 2D-FP plots illustrated in Figure 4 were constructed from the 2D-HS by considering outside and inside closest-neighbor molecules. These integrated visions on contacts were helpful in the imagining of nonpolar and polar atoms interactions contributions in the cluster lattice. The other atom…atom contact rations were resolved as H…H (60.0%). Intermolecular contacts showed the larger contribution part and H…..M (Cu and Cd) (0.0%) interactions ratio. Early studies concerned H…H connections as steric repulsive interactions that disturb the molecular system [42]. Moreover, the understanding of H…H interactions was verified and changed in the 1990s since a new type of interaction named the dihydrogen bond (DHB) was recorded in crystal structures of different organometallic complexes [42,43,44,45]. The other atom…atom intermolecular forces are illustrated in the following importance order: H….H (60.0%) > H….C(10.0%) >H….N (5.6%) >H…..O (5.40%) > H…..Cl(1.0%) >H….M(0.0%).

### 2.4. FT-IR and EDX Investigations

For the preparation of [Cl_2_Cu_4_Cd_2_(NNO)_6_(NN)_2_(NO_3_)_2_], the CH_3_CN cluster was followed up by FT-IR, as shown in Figure 5. The obtained bimetallic cluster reflected several IR bands matching with its continent functional groups. Several stretching vibrations were exposed in the cluster backbone like aliphatic and aromatic C-H, NO_3_, C=N, MeCN, C-O, C=C, Cu-O, Cd-O, Cu-N, Cd-N and M-Cl, which were cited to their expected wavenumbers (see experimental part). Kinetically, the important changes supporting the clusterization process were the vanishing of N-H (in NNH ligand) peak at 3170 cm^−1^ by the complexation NNH with metal centers and the appearing of new M-O/M-N bands at >400 cm^−1^ chemical shifts [21,22]. The broad peak at ~3400 cm^−1^ was mostly due to humidity on the crystal surface of the cluster. Peaks at ~2930–2820 cm^−1^ in both the ligand and the cluster were attributed to C-H stretching vibrations, the C=N stretching vibration in the free ligand shifted from 1610 cm^−1^ to 1560 cm^−1^ upon coordination to the metal center. The appearance of broad peaks at 265–2500 cm^−1^ in the cluster was only due to the presence of MeCN stretching vibration. The C=C and C-O stretching vibrations in both ligand and cluster were the same, with ~1480 and 1100 cm^−1^, respectively. The M-O and M-N bonds for the both metals (Cu and Cd) vibrations were in the low range ~400–450 cm^−1^.

The qualitative compositions of the cluster were confirmed by EDX analysis, as presented in Figure 6. The presence of Cu atoms was confirmed by energy signals at 1.2, 8.2 and 9.1 keV. Meanwhile, the Cd atoms peaks were cited to 3.3 and 3.5 keV positions, moreover, Cl to 2.4 keV position, respectively. Moreover, C, N and O atoms were cited to their expected atomic energy peaks as seen in Figure 6.

### 2.5. Electronic Transfer and Optical Energy Gap

The electronic absorption of the novel desired cluster and its free ligands in DMSO solvent were combined, as illustrated in Figure 7a. Peaks in the UV-region: the recorded λ_max_ = 280 nm (ε = 1.0 × 10^3^ L mol^−1^ cm^−1^) for the cluster, λ_max_ = 282 nm (ε = 1.8 × 10^3^ L mol^−1^ cm^−1^) for NNH ligand and λ_max_ = 286 nm (ε = 1.6 × 10^3^ L mol^−1^ cm^−1^) for NNOH ligand all can be assigned to π→π* ligands e-transition. The two broad and low-intensity peaks in the visible region at λ_max_ = 605 nm (ε = 2.2 × 10^2^ L mol^−1^ cm^−1^) and λ_max_ = 685 nm (ε = 8.4 × 10^2^ L mol^−1^ cm^−1^) cited to the Cu(II) d-d e-transfer in the cluster only confirmed the N-M and O-M bond coordination [46]. The experimental optical band gap energies (ΔE_g_) in DMSO were obtained by using the Tauc relation [40]. The organic ligands direct ΔE_g_ was found to be 4.18 eV, as seen in Figure 7b. Meanwhile, the metallic direct ΔEg was found to be 1.73 eV, as seen in Figure 7c. The attained ΔE_g_ results reflected the 4Cu(II) cluster centers complexes within the visible region; meanwhile, the NNO and NN ligands with invisible region electron transfer. Therefore, clusters with such optical performance properties are expected to be an important material for solar cells, optoelectronic devices and photonic devices [47].

### 2.6. Cluster Oxidation Potential toward Catecholase of Catechol

One of the goals of the present study is to evaluate the aerobic catecholase catalytic oxidation founding of the cluster. To accomplish this, 0.1 M of 3,5-di-tert-butylbenzene-1,2-diol (3,5-DBT) was mixed with 1 × 10^−4^ M of the desired cluster in DMF solvent under stirring an open RT system for around 1 h (Scheme 2). The formation of the 3,5-di-tert-butylcyclohexa-3,5-diene-1,2-dione (3,5-DTBQ) product was monitored by UV-vis analysis in 250–500 nm range [20,21,44], as seen in Figure 8.

In Figure 8a, the absorptions of 3,5-DTBQ product at λ_max_~400 nm were gradually raised by time only with the presence of the cluster catalyst. The result revealed without ambiguity that the cluster acts as an excellent oxidation catalyst since the reaction was completely finished within 1 h. Moreover, neither side products nor products were detected by the UV-vis absorption in the absence of the cluster Figure 8b.

The oxidation rates of the cluster using several concentrations of 3,5-DBT find to be suited to the Michaelis–Menten plot (Figure 9a), which was linearized to Lineweaver–Burk plot (Figure 9b), in order to figure out the Vmax, KM and Kcat kinetic parameters. The kinetic parameter values were compared to similar catalytic processes [48,49,50,51,52,53,54]. In general, the cluster catalyst results were excellent compared to other complexes, where many factors like solvents, substrate and complex nature control the ability of catalysts [2,20,51].

## 3. Materials and Methods

The commercially available CdCl_2_.4H_2_O and Cu(NO_3_)_2_.6H_2_O salts, solvents and chemicals, were served without purification. The NNOH and NNH were prepared in our lab [22]. EDX was performed using Brucker D/MAX 2500 X-ray diffractometer, with λ = 1.54 Å Cu K radiation, TU-1901 double-beam UV-visible spectrophotometer was used to measure the UV-vis. The TG was carried out using TGA SDT-Q600, FT-IR spectra were performed in the range of 4000–400 cm^−1^ of frequency using a PerkinElmer Spectrum, and HS calculation was performed using CRYSTAL EXPLORER 3.1 package [33].

### 3.1. Synthesis of [Cl_2_Cu_4_Cd_2_(NNO)_6_(NN)_2_(NO_3_)_2_].CH_3_CN Bimetallic Cluster

0.16 mmol of each salt CdCl_2_.4H_2_O and Cu(NO_3_)_2_.6H_2_O were dissolved in 50 mL of ethanol solvent. After complete dissolving, 0.16 mmol of NNOH and NNH each (in 20 mL of CH_3_CN) were added to the reaction mixture, which was stirred for 24 h in an open condition. The reaction was then stopped and subjected to the solvent-evaporation process. After 3 weeks, light green plate crystals of [Cl_2_Cu_4_Cd_2_(NNO)_6_(NN)_2_(NO_3_)_2_].CH_3_CN cluster had formed in 74% yield. Peaks of selected IR vibrations are listed as ν = 3018 (C-H), (1520–1380) (NO_3_), 2936 (C-H), 1622 (C=N), (2200–2550) (-CN), 1485 (C=C), 1020 (C-O), cm^−1^. The peak at 420–550 cm^–1^ belongs to M–O and M–N stretching vibrations [21,22]. UV-vis. (DMSO) at λ_max_ = 280, 605 and 685 nm and m.p. > 340 °C.

### 3.2. Catechol Oxidation Reaction

The catalytic part was performed using the procedure described recently in the literature [22].

### 3.3. X-Ray

Single crystal X-ray data were collected on a Bruker D8 Quest diffractometer (MoKα radiation λ = 0.71073 Å) at 298 K. The structure was solved by direct methods and refined by full-matrix least-squares methods based on F^2^ using the SHELXL software [34]. Crystal data for the desired cluster is illustrated in Table 3.

## 4. Conclusions

A new [Cl_2_Cu_4_Cd_2_(NNO)_6_(NN)_2_(NO_3_)_2_].CH_3_CN cluster of type double-open cubane core with Cd(II)-O-Cu(II) center was made available. The 3D structure of the newly synthesized cluster was proven by XRD crystal. The XRD showed the presence of both octahedral and square pyramid metal ions geometry centers. Moreover, the di-μ^2^-methoxo-trigonal pyramid-O center and tetra-μ^3^-methoxo-tetrahedral-O centers were recorded. Several shot interactions like H_CH_….O-_NO2_, C-H….π C=C_NN_ and Cl…. Π CN were detected in the lattice by XRD and computed via HS-analysis. Furthermore, the cluster composition and structural behavior were proved spectrally furthermore via FT-IR, EDX and UV-vis. Finally, the cluster recorded an excellent catecholase potential when it was applied to the Catechol O-quinone room condition oxidation reaction.

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
