# Peer review of "Synthesis of Novel Tetra(µ3-Methoxo) Bridged with [Cu(II)-O-Cd(II)] Double-Open-Cubane Cluster: XRD/HSA-Interactions, Spectral and Oxidizing Properties"

_ijms, 2020, doi:10.3390/ijms21228787_

Round 1

Reviewer 1 Report

The work presented for review needs to be worked out and completed. During the reading, the following questions and suggestions came to my mind:
- aqua-solution synthesis - no water was added in the description of the synthesis, therefore the title must be changed;
- additionally HSA means human serum albumin not Hirshfeld surface analysis mainly recorded as HS analysis - this also needs to be improved;
- What does "oxidation aspects" mean? - may be misleading may improve oxidizing properties;
- XRD study is not the same as single crystal XRD study - the term of the method used should be clarified;
- in the introduction it is necessary to emphasize the purpose of obtaining the compound and testing its individual properties, e.g. for what purpose were the thermal stability properties tested?
- What caused the protons to be detached from the ligands - which deprotonating substance was used? please pay attention to the works published by the authors in JMS 2020, 1199, 126995?
- "The cationic units of the two Cd(II) and four Cu(II) centers are not directly bonded." so are they or are they not?
- What caused the distortion of the coordination polyhedra?
- please specify the tau parameter for the square pyramid for the Cu(II) ion;
- the description under Figure 1 is incorrect, there are various things that are not in the description;
- Table 1 - please think, which bonds from the nearest surroundings should be there, I'm not convinced by throwing the bonds N1-N2 etc, why, if you don't discuss them in the text;
- CH...O type influences - the given length D-H...A, angle should be completed;
- CH...pi - does the interaction concern the center of gravity of the five membered ring? if it is a distance to the carbon atoms from the ring, it is not an interaction of pi type !

- Can we talk about interaction of Cl...pi type from the CN group? Please provide literature if we can distinguish it;
- for what purpose and what did the analysis of Hisfeld's surface prove?
- what are the H...H-type interactions?
- the description of Figure 4 is not correct;
- FTIR spectra analysis - please describe the bands precisely and analyze the change of band positions or the appearance of new ones in relation to both organic ligands and not just one;
- please provide values of epsilon factor for transitions d-d, what they prove; additionally, please register the spectra of both ligands and compare them with the spectrum of the complex;
- thermogravimetric analysis has to be refined - please show on the graph or in the table the percentage loss of mass, match the individual fragments to be separated and count the error of the whole theoretical and experimental process; the proof of the final product of decomposition has to be provided by PXRD method;
- compound as catalyst, is the process reversible and how is the catalyst recovered? is Fig. 10 b diagram correct?
- it is necessary to remove Table 2 because it has already been published in JMS 2020, 1199, 126995;
- was the synthesis conducted at room temperature?; what was the exact yield of the synthesis?
- structural data lack basic parameters like GOF, largest deep and hole, R1, R2, wR1, wR2;
- there is also a lack of CCDC number, which would indicate sending the compound to the CCDC compound database and without it the work cannot be published;

Reviewer 2 Report

This paper reports on preparation, structural characterization and reaction of Cu-Cd cluster complexes.

Significance in the field of coordination chemistry is clear, so it should be published essentially.

Before acceptance, some points should be improved or corrected.

L55, 60 Font size should be corrected.

L96 Are No.'s necessary?

L171 Are information in solid states (such as HSA) valid for 

        discussion of reaction in a solution? Otherwise, what 

       are commonly important information?

L202 cm-1 (-1 is superscript)

L212 are belongs to -> belongs to

L219 bu full-matrix -> by full matrix

L221 Show beamline used at SPring-8 or Photon Factory(?).

That's all.

Round 2

Reviewer 1 Report

The work has not been improved, I think the answer has been prepared too quickly because there is no explanation or answer to the questions and I still have some doubts and questions:
- I think that only the bonds and angles from the coordination environment of metal ions should be left in Table 1, the rest can be found in cif;
- I think that HSA is the acronym for human serum albumin and, in the case of 'Hirshfeld surface', an error in the name of the method, should be corrected, it should be marked with the HS symbol;
- please specify the tau parameter for the square pyramid for the Cu(II) ion; no answer is given;
- it is necessary to clarify the concept of centre of gravity and the effect of CH...pi;
- what are the H...H-type interactions? and what does "This is sort of Van der Waal"????? mean?
- please provide values of epsilon factor for transitions d-d, what they prove; additionally, please register the spectra of both ligands and compare them with the spectrum of the complex; the parameters of UV-Vis spectra are not discussed in the paper quoted;
- if thermogravimetric analysis of individual stages is not possible, additional analyses should be carried out using e.g. TG-MS method;
- the reaction yield has not been placed in the work;

Round 3

Reviewer 1 Report

Dear Authors,

  • please change the value for τ (tau parameter) for 5-coordinated Cu(II) center because it is parameter proposed by Addison, the value can not be 7.12 oC
  • if thermogravimetric analysis of individual stages is not possible, percentage of every step is also not present, what is the final product? - this part should be removed from manuscript;

Author Response

I would like to thank you for your effort on our MS, it is my pleasure to inform you that we have performed almost the corrections requires, moreover, all the corrections were highlighted to the text.

Please note that we have corrected the value of τ[1O2O6N1N] = 3.12 oC

The thermal part has been erased from the text.

I hope that we fulfilled most of the proposals to a degree reach up to acceptance.

Thanks